# Dopaminergic Projections from the Hypothalamic A11 Nucleus to the Spinal Trigeminal Nucleus Are Involved in Bidirectional Migraine Modulation

**DOI:** 10.3390/ijms242316876

**Published:** 2023-11-28

**Authors:** Chenhao Li, Yang Li, Wenwen Zhang, Zhenjie Ma, Shaobo Xiao, Wei Xie, Shuai Miao, Bozhi Li, Guangshuang Lu, Yingyuan Liu, Wenhao Bai, Shengyuan Yu

**Affiliations:** 1Department of Neurology, The First Medical Center of Chinese PLA General Hospital, Beijing 100853, China; lichenhao578@163.com (C.L.); li_yang1097@163.com (Y.L.); nku_zww@163.com (W.Z.); zhenjiema1218@163.com (Z.M.); xiaoshaobo6208@163.com (S.X.); xiewei_1988@163.com (W.X.); miaoshuaichn@163.com (S.M.); libozhicy@163.com (B.L.); vespa126@163.com (G.L.); liuyingyuan2023@163.com (Y.L.); wenhaobai301@163.com (W.B.); 2Medical School of Chinese PLA, Beijing 100853, China; 3School of Medicine, Nankai University, Tianjin 300071, China

**Keywords:** migraine, glyceryl trinitrate, dopamine, A11, D2 dopamine receptor, D1 dopamine receptor, GABAergic neuron

## Abstract

Clinical imaging studies have revealed that the hypothalamus is activated in migraine patients prior to the onset of and during headache and have also shown that the hypothalamus has increased functional connectivity with the spinal trigeminal nucleus. The dopaminergic system of the hypothalamus plays an important role, and the dopamine-rich A11 nucleus may play an important role in migraine pathogenesis. We used intraperitoneal injections of glyceryl trinitrate to establish a model of acute migraine attack and chronicity in mice, which was verified by photophobia experiments and von Frey experiments. We explored the A11 nucleus and its downstream pathway using immunohistochemical staining and neuronal tracing techniques. During acute migraine attack and chronification, c-fos expression in GABAergic neurons in the A11 nucleus was significantly increased, and inhibition of DA neurons was achieved by binding to GABA A-type receptors on the surface of dopaminergic neurons in the A11 nucleus. However, the expression of tyrosine hydroxylase and glutamic acid decarboxylase proteins in the A11 nucleus of the hypothalamus did not change significantly. Specific destruction of dopaminergic neurons in the A11 nucleus of mice resulted in severe nociceptive sensitization and photophobic behavior. The expression levels of the D1 dopamine receptor and D2 dopamine receptor in the caudal part of the spinal trigeminal nucleus candalis of the chronic migraine model were increased. Skin nociceptive sensitization of mice was slowed by activation of the D2 dopamine receptor in SP5C, and activation of the D1 dopamine receptor reversed this behavioral change. GABAergic neurons in the A11 nucleus were activated and exerted postsynaptic inhibitory effects, which led to a decrease in the amount of DA secreted by the A11 nucleus in the spinal trigeminal nucleus candalis. The reduced DA bound preferentially to the D2 dopamine receptor, thus exerting a defensive effect against headache.

## 1. Introduction

Migraine is a primary headache caused by brain dysfunction, often presenting as a unilateral throbbing headache with nausea, vomiting, photophobia, phonophobia, and cephalic and facial nociceptive hypersensitivity, with anxiety and depression often present as comorbidities [1]. Global epidemiologic surveys show that the annual incidence of migraine is 12%, and according to the Global Burden of Disease Study published in 2018, migraine has the second highest rate of lifetime disability among all diseases; it is the leading cause of lifelong disability in the 15–49 age group. Migraine has become a global public health problem, with serious medical and economic burdens on patients, families, and society [2,3]. Migraine is more than just a headache. It is now recognized as a complex neurological disorder affecting multiple cortical, subcortical, and brainstem regions that regulate autonomic, emotional, cognitive, and sensory functions.

As a migraine progresses, it can be accompanied by a variety of autonomic symptoms (nausea, vomiting, sinus congestion, nasal leakage, tearing, ptosis, yawning, frequent urination, and diarrhea), affective symptoms (depression and irritability), cognitive symptoms (difficulty concentrating, difficulty speaking, transient amnesia, and decreased ability to navigate in familiar environments), and sensory symptoms (photophobia, phonophobia, muscular tenderness, and skin tenderness). These clinical features point to the hypothalamus as a prodromal symptom/attack initiating brain structure. Some features pointing to this role include hypothalamic involvement in the regulation of yawning [4], fatigue, and mood changes [5]; the circadian rhythmicity of attacks [6]; and the relationship between attacks and hormonal status and the menstrual cycle [7]. The hypothalamus also has various neuroanatomical connections with the pain-modulating system and the spinal trigeminal nucleus [8,9]. Previous clinical imaging studies in migraine patients have also suggested that the hypothalamus plays a crucial role in the pathophysiology of migraine chronicity and the acute pain phase of migraine. While the posterior portion of the hypothalamus appears to be important for the acute pain phase, the anterior portion appears to play an important role in migraine chronification and attack [10,11,12]. 

The orexin system and dopaminergic system of the hypothalamus may play important roles in migraine and changes in migraine auras [13,14]. Among them, the dopamine system plays a crucial role in a variety of physiological functions, such as reward, motor function, motivation, goal-directed behavior, memory, learning, and pain [15,16,17]. Typical aura symptoms, such as yawning and fatigue, as well as changes in appetite and nausea, involve the dopaminergic system [13,14]. Interestingly, intravenous injection of dopamine agonists can induce common aural symptoms, such as nausea, vomiting, and yawning, several hours before migraine attack [18,19]. Dopaminergic agonists such as apomorphine increase yawning, dizziness, nausea, and vomiting in migraineurs [13,18,20,21]. It has been suggested that the midbrain dopaminergic system may play a role in cluster headache pathophysiology [22]. Dopamine antagonists such as metoclopramide, commonly used to treat nausea in migraine, have been shown to be effective in the treatment of migraine itself [23,24,25,26,27,28]. The above research results show that the dopaminergic system is closely related to the pathophysiological process of migraine [13,29].

Dopaminergic neurons are enriched in several brain regions designated A8–A16 [15]. Among them, the A11 dopaminergic system is the only dopaminergic system that projects downward to the medullary dorsal horn (MDH) and spinal dorsal horn (SDH) and is the only source of dopamine in the MDH and SDH [30,31]. Dopamine receptors are expressed in primary nociceptors and spinal neurons in different layers of the dorsal horn of the spinal cord, indicating that dopamine can regulate pain signals by acting on presynaptic and postsynaptic targets. Therefore, we suspect that dopamine projection from the A11 nucleus descending to spinal trigeminal nucleus candalis (Sp5C or TNC) may be involved in pain regulation in migraine [32].

In this study, we used the GTN-induced mouse model as the research object to explore the role of the A11 nucleus in acute migraine attack and the chronic process of headache.

## 2. Results

### 2.1. Activation of GABAergic Neurons in the A11 Nucleus after GTN Intervention and Inhibition of Dopaminergic Neurons

c-fos is recognized as a marker of neuronal activation after injurious stimulation, and its expression is upregulated in response to either acute or chronic intervention with GTN [33,34]. Immunofluorescence staining showed that acute or chronic stimulation with GTN caused a significant increase in c-fos expression in the hypothalamic A11 nucleus in the GTN-induced mouse migraine model (control vs. AM vs. CM = 2.7 ± 2.36 vs. 41.44 ± 10.1 vs. 50. 63 ± 6.44, Figure 1B,C). To clarify the nature of activated neurons in the A11 nucleus, we used a variety of neuron-specific markers, such as vGluT2 (glutamatergic neuron-specific marker), TH (catecholaminergic neuron-specific marker), GAD65 + 67 (GABAergic neuron-specific marker), and TPH2 (pentraxin neuron-specific marker), along with other neuronal-specific markers, with c-fos determined by immunofluorescence double-label staining. We found that in the GTN-induced mouse migraine model, the activated neurons in the A11 nucleus were GABAergic neurons (Figure 1D), the A11 nucleus was mainly composed of dopaminergic neurons and GABAergic neurons, and no other neurons were found.

Previous studies have shown that dopaminergic neurons in the A11 nucleus are involved in the regulation of neuropathic pain through downstream inhibitory pathways toward the TNC [35], and we suspected that GABA can act as an intermediate neuron to modulate dopaminergic neurons in the A11 nucleus via paracrine regulation. We found that A11 nucleus dopaminergic neurons colocalized with GABA A receptors through immunofluorescence staining (Figure 1E). Additionally, the proportion of colocalization was higher in the acute and chronic groups compared to the control group, and there was an enrichment of GABAA receptors on the surface of DA neurons in the A11 nucleus after GTN intervention (Appendix A). This alteration was not present in other cells within the A11 nucleus. The above results suggest that, to a large extent, A11 nucleus GABAergic neurons inhibit dopaminergic neurons by paracrine GABA transmitters binding to GABA A-type receptors on the surface of dopaminergic neurons. To investigate whether the metabolic levels of DA neurons and GABAergic neurons were altered as a result of GTN stimulation, we analyzed changes in the expression of TH and GAD in the A11 nucleus of mice before and after NTG intervention by protein blotting analysis (Figure 1F). The results showed that acute and chronic stimulation by NTG did not change the expression of TH and GAD, i.e., dopaminergic neurons and GABAergic neurons did not undergo changes in metabolic levels.

### 2.2. Specific Ablation of Dopaminergic Neurons in the A11 Nucleus Leads to More Intense Allodynia and Photophobic Behavior in Mice after GTN Stimulation

To investigate the role of dopaminergic neurons in the A11 nucleus in the NTG-mediated mouse migraine model, we injected 6-OHDAn into the bilateral A11 nucleus of mice using a stereotaxic technique to specifically damage the dopaminergic neurons (Figure 2B) and tested the mechanical pain thresholds of the head, face, and hind paw in model mice before and after the damage. To better investigate the effects of acute GTN intervention versus chronic GTN intervention on model mice in the presence of specific destruction of DA neurons in A11, we divided acute GTN-treated mice and chronic GTN-treated mice into two different groups. For the acute intervention mouse model, we took the pain threshold before intraperitoneal injection of GTN as the baseline and recorded the pain threshold 1 h, 2 h, 3 h, 6 h, 12 h, 24 h, and 48 h after injection. There was no significant difference in the baseline pain threshold of the GTN + lesion group compared to the GTN + VEH group. At 1 h, 6 h, 12 h, 24 h, and 48 h after intraperitoneal injection, the plantar pain threshold of the GTN + lesion group was significantly lower than that of the GTN + VEH group, whereas the cephalic pain threshold of the GTN + lesion group was significantly lower than that of the GTN + VEH group at 2 h, 6 h, 12 h, 24 h, and 48 h. For the chronic GTN intervention mouse model, we performed cephalofacial and plantar mechanical pain threshold measurements before each intraperitoneal administration of GTN. We found that the GTN + Sham group and GTN + lesion group exhibited progressive and persistent basal allodynia after chronic GTN intervention. The basal pain threshold decreased more rapidly in the GTN + lesion group than in the GTN + sham group. The GTN + lesion group had lower basal pain thresholds in the hind paws than the GTN + Sham group on Days 3 and 5 (Figure 2E) and in the head on Days 3, 7, and 9.

In addition to mechanical pain sensitization of the head, face, and hind paws, we used a modified light/dark box to assess light aversion behavior (photophobia). According to the results of our team’s previous experiments, during the first 50 min after GTN injection, mice showed a significant reduction in locomotor activity due to the cardiovascular effects (hypotension) of GTN. As a result, these mice were inactive in the box for more than 90% of the time. Previous studies have shown that GTN model mice exhibit photophobic behavior 1 h after intraperitoneal injection of GTN, which is mainly reflected by the time spent in the bright box and the number of jumps between the bright and dark boxes. Therefore, we performed photophobia experimental measurements 1 h after GTN administration. For both acute and chronic GTN-treated mice, the GTN + lesion group showed more significant photophobic behavior than the other three groups (Figure 2D,F, acute: VEH + Sham vs. GTN + Sham vs. VEH + lesion vs. GTN + lesion = 213.12 ± 35.56 vs. 172.37 ± 54.55 vs. 187.69 ± 60.94 vs. 100.91 ± 39.67; Figure 2F, chronic: VEH + Sham vs. GTN + Sham vs. VEH + lesion vs. GTN + lesion = 210.22 ± 26.59 vs. 197.65 ± 49.04 vs. 188.88 ± 51.78 vs. 112.11 ± 50.91).

### 2.3. Dopaminergic Neurons in the A11 Nucleus Produce Projections to SP5C, and DA Secreted from the Terminal Endings Binds to D1DR and D2DR Receptors to Play a Role

Nociceptive sensitization and photophobic behavior due to NTG stimulation were further exacerbated by bilateral destruction of dopaminergic neurons specifically in the A11 nucleus. This behavioral alteration was associated with the disruption of dopaminergic inputs in brain regions receiving projections from A11 nucleus accumbens dopaminergic neurons. As previously studied, the A11 nucleus, as the sole source of dorsolateral medullary DA, emits dopaminergic projections to SP5C involved in the modulation of neuropathic pain. It is therefore reasonable to speculate that dopaminergic neurons of the A11 nucleus emit projections to SP5C, which is involved in the modulation of migraine-related nociceptive sensitization. We validated the above pathways using neuronal tracing techniques. We injected the cis-tracer virus AAV9: pAAV-EF1-DIO-EGFP-WPRE into the bilateral A11 nucleus of TH-cre mice, and after waiting 21 days for viral expression, mouse tissues were subjected to staining, and we found a large number of projection fibers in the bilateral SP5C region (Figure 3A,B). We also found A11 projection fibers in the SP5C region wrapped around D1DR- and D2DR-positive neurons with projection fibers (Figure 3A,B). We also performed reverse validation of the above pathway using FG and the retrograde tracer virus retro: pAAV-hSyn-DIO-mCherry-WPRE (Figure 3C,D), and the conclusion remained reliable. This suggests that dopaminergic neurons of the A11 nucleus are greatly involved in the regulation of nociceptive sensitization in migraine by sending projections to SP5C and secreting DA to bind to downstream D1DR and D2DR.

### 2.4. D2DR in the SP5C Region of Mice Is Predominantly Distributed in VGAT+ Neurons, Whereas D1DR Is Predominantly Distributed in GLU+ Neurons

To clarify the distribution characteristics of D1DR and D2DR in SP5C, we used transgenic mice combined with immunofluorescence staining. By multiple immunofluorescence staining, we found that in the SP5C region, D2DR+ neurons were mainly distributed in the deep layer of SP5C, which was basically the same as the region of VGAT+ neurons. D1DR+ neurons were mainly distributed in the shallow layer of SP5C (near SP5), which was roughly similar to the distribution range of GLU+ neurons (Figure 4A). By multiple immunofluorescence staining, we found that most D2DR+ neurons colocalized with VGAT+ neurons and that most D1DR+ neurons colocalized with GLU+ neurons, but a few D1DR+ neurons near the deeper layer still colocalized with VGAT+ neurons (Figure 4B). This fraction of neurons was most likely D1DR+/D2DR+ dual immunoreactive GABA neurons (Figure 4C).

### 2.5. Inhibition of D2DR in the SP5C Region of Mice Enhances Allodynia, Whereas Inhibition of D1DR Attenuates Allodynia

To clarify the role played by DA receptors downstream of A11 projections in cutaneous pain sensitization in migraine, we cannulated unilateral SP5C in mice and administered the D2DR antagonist spiperone and the D1DR antagonist SCH23390 by microinjection. We found that cephalalgia thresholds and hind paw mechanical pain thresholds on the same side as spiperone administration in mice tended to decrease compared with the control group, whereas the opposite was true for mice given SCH23390. Mice given SCH23390 had a significant increase in cephalofacial pain threshold as well as hind paw mechanical pain threshold compared to the GTN + VEH group, but these remained lower than in the VEH group (Figure 5A).

**Figure 4 ijms-24-16876-f004:**
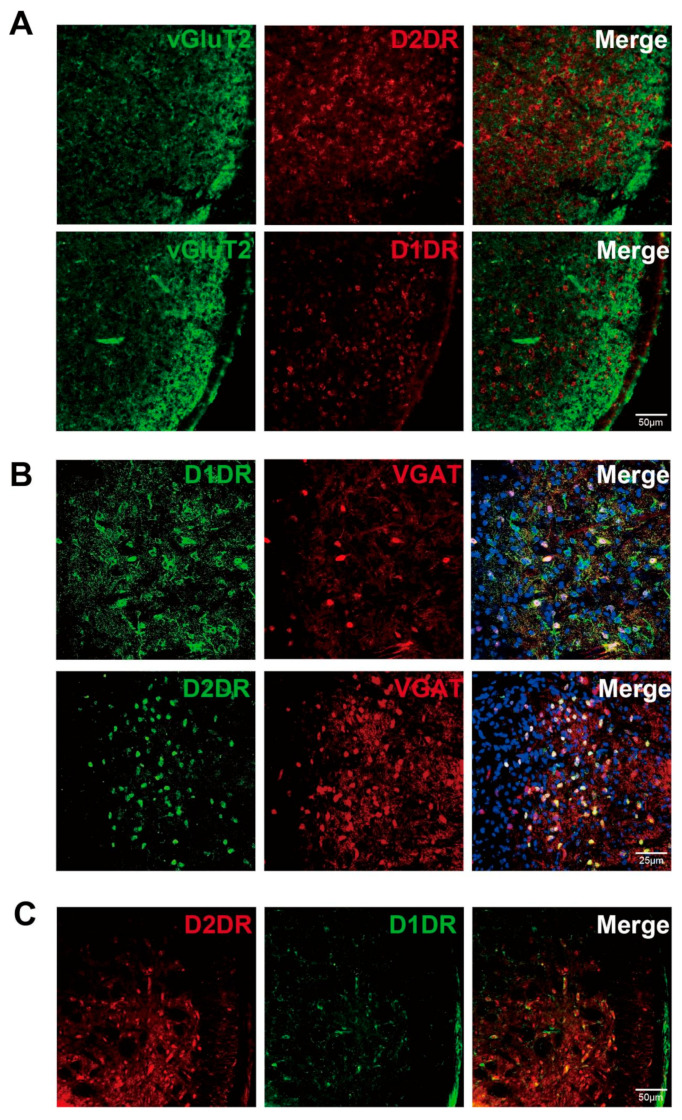
Distribution characteristics of D1DR and D2DR in SP5C. (**A**) Most of the D2DR+ (red) cells are distributed in the deep layer of SP5C, and a small number of D2DR+ (red) cells are distributed in the shallow layer colocalized with vGluT2+ (green) cells. The majority of D1DR+ (red) cells are distributed in the shallow layer of SP5C colocalized with vGluT2+ (green) cells. (**B**) A small number of D1DR+ (green) cells colocalize with VGAT+ (red) cells, and the vast majority of D2DR+ (green) cells colocalize with VGAT+ (red) cells. (**C**) Some cells distributed in the shallow layer of SP5C exhibit D1DR+ (green) and D2DR+ (red) double immunoreactivity positivity.

### 2.6. Elevated D2DR Receptor Expression in Mouse SP5C during Migraine Chronicity

To explore whether expression of the SP5C receptor is altered during acute migraine attack and chronicity, we used WB to quantitatively analyze the protein expression of D1DR and D2DR in SP5C of model mice. The results showed that after the mice received chronic GTN stimulation, the D2DR expression of bilateral SP5C was significantly higher than that of the control group, and D1DR showed a trend of elevated expression, but there was no significant difference (Figure 5B). In contrast, there was no significant change in D1DR and D2DR protein expression of SP5C in mice under acute GTN stimulation conditions.

## 3. Discussion

In the present study, we revealed that (1) GABAergic neurons in the hypothalamic A11 nucleus are activated in a nitroglycerin-mediated mouse migraine model, and secreted GABA binds to GABA A-type receptors on the surface of dopaminergic neurons in the A11 nucleus and exerts postsynaptic inhibitory effects; (2) there is a dopaminergic-specific projection of dopaminergic neurons in the A11 nucleus to the SP5C region; (3) specific modulation of DA receptors D1DR and D2DR in the SP5C region exerts opposing modulatory effects on GTN-induced migraine-associated cutaneous nociception; and (4) during migraine chronicity, D2DR expression in SP5C of mice is upregulated for adaptive chronic pain stimuli. In conclusion, our results suggest that downstream dopaminergic projections from the hypothalamic A11 nucleus to SP5C are involved in the modulation of GTN-mediated migraine and that this pathway functions in pathological states.

In the past, it was thought that all sensory and injurious inputs received by the hypothalamus [36,37,38,39] originated in brainstem nuclei such as the parabrachial nucleus [40] and periaqueductal gray matter [41,42]. It is now widely known that neurons in these hypothalamic regions and nuclei receive large direct injurious inputs via the trigeminal hypothalamic tract (THT), a bundle of neurons originating in the spinal trigeminal nucleus that processes sensory signals from all organs and tissues of the head, including the meninges [43,44,45,46]. Axons of persistently sensitive THT neurons project directly to the hypothalamic nucleus and may be involved in pain, temperature, feeding, and autonomic and neuroendocrine signaling [47,48,49]. Some relevant neuroimaging studies have reported increased functional connectivity between the anterior hypothalamus and the trigeminal spinal nucleus during the preictal phase in patients with episodic migraine and significant connectivity between the anterior hypothalamus and the spinal trigeminal nucleus in the chronic migraine group, whereas there was no connectivity between these two regions during the interictal phase [11]. Neuroimaging studies using PET or functional magnetic resonance imaging have revealed dorsal pontine and hypothalamic activation during migraine attacks [12,50,51,52,53]. Recent studies have shown that the hypothalamus is activated in migraineurs during the preictal phase of triggered attacks [54] and spontaneous attacks [11,55]. These results suggest that the hypothalamic nucleus is activated during the preictal phase of migraine attacks and increases functional connectivity with the spinal trigeminal nucleus, lowering the threshold for activation of the spinal trigeminal nucleus by stimulation. The hypothalamic A11 nucleus, as the only dopaminergic input to the spinal trigeminal nucleus, is likely to play a facilitating or inhibitory role in migraine by interfering with the TNC during the pre-migraine attack and headache phases.

Previous studies have demonstrated that the downstream dopaminergic pathway of the hypothalamic A11 nucleus contributes to pain modulation [40,56,57,58,59] and that the downstream pain-modulating pathway originating from the A11 nucleus plays an important role in spinal dorsal horn neuron-induced pathological pain plasticity. The downstream pain-modulating pathway originating from the A11 nucleus has also been shown to be involved in the modulation of formalin-induced trigeminal pain behavior [40,56], and it has also been shown to be involved in the modulation of neuropathic pain induced by orbital nerve ligation [60]. However, it was not clear whether the A11 nucleus played a role in the GTN-induced migraine model in mice. First, by immunofluorescence staining, we found that a significant increase in the number of c-fos+ cells in the region of the A11 nucleus was observed in both acute GTN-treated and chronic GTN-treated model mice, suggesting that neurons in the A11 nucleus are involved in the pathological process of migraine. Thus, we explored the composition of the A11 nucleus, and by immunofluorescence staining, we found that the A11 nucleus was mainly composed of GABAergic neurons and dopaminergic neurons. Curiously, in the GTN-induced migraine model, GABAergic neurons but not DA neurons were activated. Since GABAergic neurons are mostly interneurons, they usually secrete GABA transmitters that act on neighboring neurons or glia via paracrine secretion. Therefore, we hypothesized that GABAergic neurons in the A11 nucleus were activated during migraine attack and secreted GABA transmitters that acted on dopaminergic neurons in the A11 nucleus, and dopaminergic neurons were inhibited and therefore did not show an increase in c-fos expression [61,62]. This is consistent with trends in clinical imaging studies in migraine patients. Endogenous DA is reduced in migraine patients during attacks as well as during chronic relapses [63]. To verify this hypothesis, we found that dopaminergic neurons in the A11 nucleus colocalized with GABA type A receptors by immunofluorescence staining. The GABAA receptor opened chloride channels on the surface of neurons during excitation, and inward chloride flow hyperpolarized the postsynaptic membrane, which induced a fast inhibitory postsynaptic potential and inhibited dopaminergic neurons in A11. This suggests that it is highly likely that dopaminergic neurons in the A11 nucleus play a role in GTN-induced migraine. Previous studies injecting GABAA receptor agonists into the rat A11 nucleus have found that GABAA receptor activation in the A11 nucleus attenuates mechanical hypersensitivity to facial formalin injection [56] as well as after orbital nerve ligation [64], which is in line with our study.

To further validate this, we used 6-OHDA to specifically damage dopaminergic neurons in the bilateral A11 nucleus of mice, and we found that mice with dopaminergic neurons in A11 damaged exhibited lower head-face pain thresholds and hind paw pain thresholds under acute GTN stimulation than the undamaged group, and the damaged group was nociceptively sensitive to pain for a longer period after acute GTN stimulation. In contrast, under chronic GTN intervention, the damaged group showed lower and more rapidly decreasing baseline head-face and hind paw pain thresholds. In the pain-related photophobic behavior assay, the damaged group showed significant aggravation of photophobic behavior 1 h after GTN (both acute and chronic) intervention. This suggests that dopaminergic neurons in A11 play a role in GTN-mediated nociceptive sensitization and photophobic behavior in the migraine model, which is similar to the results in a neuropathic pain model after orbital nerve ligation [35,64].To clarify in which brain regions the dopaminergic projections from the A11 nucleus regulate migraine pathogenesis, we used a cis-tracer virus in combination with TH-cre mice to explore the projection sites of the A11 nucleus. We found dopaminergic projections from the A11 nucleus in several brain regions, including SP5C, LPMR, LPLR, APTD, MHb, and LHb. Previous studies have shown that dopaminergic neurons in the A11 nucleus are the sole source of DA in the MDH and SDH [56] and are involved in the regulation of neuropathic pain via D1-like receptors and D2-like receptors. Activation of D1DR promotes neuropathic pain, whereas activation of D2DR exerts some inhibitory effects on neuropathic pain. To clarify whether D1-like receptors and D2-like receptors play the same role in the migraine model, we performed immunofluorescence staining of SP5C in mice. By staining, we found that D2DR was mostly distributed in the deep layer of SP5C and colocalized with VGAT+ neurons. D1DR was mostly distributed in the superficial layer of SP5C and colocalized with VGLUT2+ neurons. However, there were still some D1DRs close to the deep layer of SP5C and colocalized with VGAT+ neurons, and we speculate that the surface of these neurons may be GABAergic neurons colocalized with D1DR and D2DR receptors. We observed dopaminergic fibers projecting to SP5C through the A11 nucleus. Dopaminergic neurons were distributed in both the deep and superficial layers of SP5C but were relatively densely distributed in the superficial layer. Colocalization of projecting fibers with D1DR+ neurons and D2DR+ neurons was observed. This lays an anatomical foundation for the regulation of migraine by the A11 downstream DA pathway. We injected the D1DR antagonist SCH23390 with the D2DR antagonist spiperone into mouse SP5C and found that specific antagonism of D1DR attenuated GTN-mediated cephalo-facial and hind paw nociceptive sensitization, whereas specific antagonism of D2DR exacerbated GTN-mediated cephalo-facial and hind paw nociceptive sensitization. This is consistent with trends in the modulation patterns of the A11 nucleus in peripheral nerve pain as well as neuropathic pain in previous studies [35,40,64].

Notably, pain sensitization in GTN model mice was more severe when we used 6-OHDA to specifically damage dopaminergic neurons in the bilateral A11 nucleus of mice. However, during the formation of the migraine model, GABA neurons of the A11 nucleus were activated, inhibiting dopaminergic neurons within the nucleus and reducing downstream DA release yet providing pain suppression. We believe that this unique mode of regulation is largely related to the differential expression of D1DR and D2DR in SP5C and the dose dependence of D1DR and D2DR receptors on DA. Both D1DR and D2DR receptors in SP5C function under normal physiological conditions. In contrast, when mice were stimulated with GTN, GABA neurons in the A11 nucleus were activated and inhibited dopaminergic neurons, resulting in a decrease in DA content in SP5C. Since DA has a greater affinity for the D2DR receptor than the D1DR receptor [65,66], when DA levels are decreased, DA preferentially binds to D2DR and thus negatively modulates headache, which leads to some degree of headache relief. In contrast, when mice were subjected to bilateral dopaminergic neuron destruction, the DA content of SP5C became minimal and could not effectively activate D2DR, so headache was exacerbated. This needs to be verified using transsynaptic viruses in combination with chemogenetic or optogenetic viruses to more finely modulate downstream neurons.

Previous studies have shown that DA receptor expression levels are influenced by certain microRNAs [67]. Early life stress enhances the susceptibility to late life stress through decreasing microRNA-9 expression and subsequently upregulating dopamine receptor D2 expression in the nucleus accumbens [68]. microRNA-326 may be a novel target of escitalopram. In a chronic migraine model, it has also been found that upregulation of miR-155-5p expression in the TNC inhibits silent information regulator 1 and enhances the inflammatory response [69]. The above results give us some important inspiration. According to our WB results, the expression of D2DR in SP5C is upregulated during headache chronicity in mice, while the expression of D1DR does not change significantly, which is precisely an adaptation to chronic pain, and D2DR is upregulated to resist the stimulation of pain in mice. Perhaps the alteration of DA receptor levels in SP5C of mice after chronic GTN intervention is also influenced by an unknown microRNA. This needs to be explored further.

Some studies have found that the A11 nucleus contains a small number of CGRP+ neurons in addition to DA neurons and GABA neurons, and that these neurons are dual TH+/CGRP+ immunoreactive neurons [13,56]. These neurons are predominantly located on the caudal side of the A11 nucleus [70], and no study has yet explored the role of these dual-immunopositive neurons in migraine. Unraveling the mechanism of influence of dual TH+/CGRP+ immunoreactive neurons in the A11 nucleus is significant for understanding headache transmission.

## 4. Materials and Methods

### 4.1. Animals

Specific pathogen-free (SPF) C57BL/6 J mice (males, 20 to 30 g, 8–10 weeks old, SiPeiFu Biotechnology Co., Ltd., Beijing, China) were housed in a controlled environment (23 ± 2 °C, 55 ± 15% relative humidity, 12:12 h light/dark cycle). Standard diet and tap water were available ad libitum. The sample size was calculated by G*Power (ver. 3.1.9.7) based on the repeated measures design (power = 0.85). The experimental procedures were approved by the Institutional Animal Care and Use Committee, Chinese People’s Liberation Army (PLA) General Hospital, following the Regulations for the Administration of Affairs Concerning Experimental Animals.

We used the up–down method, where, if the animal produced a negative response, the researchers applied the filament with the next-greater force. If the animal produced a positive response, the stimulus was decreased. After the first breaking point (change in response), four more stimulations were applied, the response pattern and final filament were noted, and 50% withdrawal thresholds were calculated using a freely available online algorithm at https://bioapps.shinyapps.io/von_frey_app/ (accessed on 3 October 2023) [71].

### 4.2. Migraine Induction

Glyceryl trinitrate (GTN) was prepared from a stock solution of 5.0 mg/mL glyceryl trinitrate in 30% alcohol, 30% propylene glycol, and water (Beijing Yimin Pharmaceutical Co., Ltd., Beijing, China). GTN was freshly diluted with 0.9% saline to a solution with a concentration of 0.5 mg/mL (10 times dilution). The diluted GTN was stored in a glass bottle at 4 °C away from light to ensure that it was ready for use. The injection volume of C57 mice was 10 mg/kg, which was converted to 20 μL/g [72]. The mice were divided into three groups according to the mode of administration: control group, acute group, and chronic group. The control group was injected with vehicle every other day for a total of five injections. The acute group was injected intraperitoneally with the drug every other day for a total of five injections. The first four injections were with vehicle and the last was with GTN. The chronic group was injected with GTN every other day for a total of 5 injections (Figure 1A).

### 4.3. Cannula Implantation and Drug Microinjection

The mice were fasted from food and water for 12 h before stereotaxic surgery, and after induction of anesthesia with 4% isoflurane by the small animal anesthesia machine, 3% sodium pentobarbital solution (0.22 mL/100 g) was injected intraperitoneally for deep anesthesia, and then the mice were fixed in the stereotaxic device (69105; RWD Life Science, Shenzhen, China). After skin preparation and sterilization of the mouse head, the skull was fully exposed, cleaned of surface connective tissue, and cranially drilled using a 0.5 mm dental drill. Next, guide cannulas (27 gauge, 3.5 mm/6.0 mm base, 62004; RWD Life Science, Shenzhen, China) were implanted bilaterally into SP5C (18° angle at coordinates anteroposterior [AP] −6.15 mm, medio-lateral [ML] ±1.60 mm, dorsoventral [DV] −5.30 mm). The cannula was secured to the skull with screws with dental cement, and a pin core was inserted to avoid cannula obstruction, ensuring that the mice that underwent surgery recovered for at least 10 days before performing behavioral tests.

Mice were lightly maintained under anesthesia with low concentrations of isoflurane (1–1.5%), the needle core was withdrawn, the injection cannula was slowly inserted (#33, 62204; RWD Life Science, Shenzhen, China), and the injection cannula was connected to a 10 μL microsyringe via a syringe pump-driven polyethylene tube (Hamilton, Reno, NV, USA) (788130; KD Scientific, Holliston, MA, USA). D1R receptor antagonist (SCH23390; Sigma, St. Louis, MO, USA), D2R receptor antagonist (spiperone, Sigma), or vehicle (10% DMSO, 40% PEG300, and 5% Tween 80) was injected at a flow rate of 50 nL/min (200 nL/side) into SP5C, and the injection cannula was left in place for 5 min after injection to allow drug diffusion [35]. After the injection was completed, the injection cannula was removed, and the needle core was reinserted. Before sampling, mice were injected with Evans Blue (E2129; Sigma, St. Louis, MO, USA) into the bilateral cannula and observed under a light microscope (DP73; Olympus, Tokyo, Japan) to verify the accurate placement of the cannula. Cannula administration was usually performed 30–40 min prior to behavioral measurements.

### 4.4. Chemical Lesion of A11 Dopaminergic Neurons

Mice were first injected with desipramine (25 mg/kg, i.p., Sigma) to protect noradrenergic neurons [40]. Next, the mice were anesthetized with 3% pentobarbital sodium and placed in a stereotaxic frame. Using a sterile technique, a 10 μL microinjector (Hamilton) was used to inject 1 μL of 6-hydroxydopamine (6-OHDA) (2 μg/μL, MCE, Romulus, MI, USA) unilaterally into the A11 nucleus (AP, −2.30 mm; ML, 0.25 mm; DV, 4.0 mm). The needle was left in place for an additional 10 min before withdrawal. The chemical lesion-produced ablation of A11 dopaminergic neurons was examined by immunohistochemical staining with an antibody against TH (a biomarker for dopaminergic neurons) [35].

### 4.5. Abnormal Skin Pain Sensitization Behavior

The degree of skin pain sensitivity of mice was evaluated by measuring the mechanical pain threshold around the orbit and hind paw [73,74]. Before measuring the mechanical pain threshold of the hind paw, the mice were placed in a plexiglass box with a dense grid at the bottom to adapt for 30 min. The mechanical pain threshold was measured by placing the von Frey filament vertically on the mouse hind paw and bending it with force. The von Frey filament was bent for 3 s or until a positive reaction was induced (the mice showed rapid paw retraction, licking, or hind paw lifting without dropping), and the results were recorded. Before measurement of the periorbital mechanical pain threshold, the mice were placed on a 5 × 5 cm platform 30 cm from the tabletop to adapt for 20 min, and the pain threshold was also measured with a von Frey filament. A positive response was considered when the animal quickly retracted its head or rubbed its face in the measurement of mechanical pain threshold in the periorbital region. The mechanical pain threshold was defined as the force that would first evoke a positive response three times in five or more trials. The vFF was calibrated by the manufacturer. The force ranges of the vFF applied to the periorbital region and the hind paw plantar were 0.008–0.4 g (0.008, 0.02, 0.04, 0.07, 0.16, and 0.4 g) and 0.07–4 g (0.07, 0.16, 0.4, 0.6, 1.0, 1.4, 2.0, and 4.0 g), respectively. The pain threshold of the GTN-induced acute migraine model group was measured 1.5–2 h after GTN injection, and that of the chronic migraine group was measured before each GTN injection [71].

### 4.6. Light/Dark Test

For the behavioral assays, all mice were acclimated in the environment for at least 2 h to exclude the effect of sudden environmental changes on mouse behavior. We used a modified light/dark box with infrared beam tracking (XR-XB120; Shanghai Xinruan Information Technology Co., Ltd., Shanghai, China) to detect photophobia (photophobia is an important feature of migraine) in the GTN-induced mouse migraine model. The photophobia assay device consisted of two boxes of the same size (30 cm wide × 30 cm deep × 30 cm high), one painted white and brightly lit (1000 lx) with an LED panel, and the other painted black and not lit (<5 lx). A corridor (7 cm × 7 cm) connected the two compartments and allowed the mice to move freely. Photophobic behavior was measured 110–120 min after VEH/GTN intraperitoneal injection of the mice and recorded for a total of 10 min. Each mouse was gently placed in the center of the bright box with its back to the dark area and then the behavior was recorded. In the interval between animal changes, the equipment was cleaned with 75% alcohol and left to evaporate.

### 4.7. Antegrade and Retrograde Tracing

Mice were anesthetized with 3% pentobarbital sodium and placed in a stereotaxic frame. A total of 200 nL of the cybernetic tracer virus AAV9: pAAV-EF1-DIO-EGFP-WPRE was injected with a 10 μL microinjector into the A11 nucleus (AP, −2.3 mm; ML, 0.25 mm; DV, 4.0 mm), and the retrograde tracer fluorogold (FG) (3%, Thermo Fisher, Waltham, MA, USA) and retrograde tracer virus retro: pAAV-hSyn-DIO-mCherry-WPRE were injected into SP5C (18° angle at coordinates anteroposterior [AP]-6.15 mm, mediolateral [ML] ± 1.60 mm, dorsoventral [DV] −5.30 mm). The needle was left in the corresponding area for an additional 15 min before withdrawal from the brain. Twenty-one days after tracer virus stereotactic injection (5–7 days after FG stereotactic injections), the mice were anesthetized with pentobarbital sodium and perfused with 4% paraformaldehyde. The specimens were sectioned for direct observation or further immunofluorescence staining [35].

### 4.8. Immunofluorescence Staining

The mouse brain was fixed with 4% paraformaldehyde for 12 h and then dehydrated in turn in 15% sucrose solution and 30% sucrose solution. After embedding with OCT glue, the mouse brain was sliced into 30 μm tissue sections (Po/LP: bregma −1.70–2.70 mm, A11: bregma −2.18–2.46 mm, TNC: bregma −7.76–8.24 mm) using a frozen microtome (Leica 1950 M). The tissue sections were sealed with blocking buffer (10% normal goat serum, 0.5% Triton X-100, dissolved in 0.1 M PBS) at room temperature (RT) for 1 h. Then, the sections were incubated with primary antibody at 4 °C for 12 h. After cleaning with 0.1 M PBS three times, the sections were incubated with secondary antibody diluted in secondary antibody dilution buffer for 2 h at RT. The species and dilutions of all primary and secondary antibodies are shown in Table 1. Then, the sections were rinsed in 0.1 M PBS three times, mounted with antifade mounting medium, and stained with 4′,6-diamidino-2-phenylindole (DAPI, P0131, Beyotime, Shanghai, China). Magnified images (×20 objective) were captured under a fluorescence microscope (BX43, Olympus, Tokyo, Japan) using cellSens standard software (version 1.18, Olympus). We used ImageJ software (version 1.52p, National Institutes of Health, Bethesda, MD, USA) to measure the immunofluorescence intensity and GraphPad Prism 9.0 to perform the statistical analysis [71].

### 4.9. Western Blot Analysis

Western blot analysis was used to detect the metabolic levels of the A11 nucleus and expression levels of dopamine receptors (D1DR and D2DR) in SP5C during migraine attack and chronicity. After the mice were anesthetized with 3% pentobarbital sodium and perfused with 0.1 M PBS, the brains of the mice were placed on ice, and the target brain area was quickly removed and temporarily stored in liquid nitrogen.

The target brain area was weighed and placed in radioimmunoprecipitation (RIPA) lysis buffer (P0013B, Beyotime) containing phenylmethylsulfonyl fluoride (PMSF, ST506, Beyotime) and protease phosphatase inhibitor mixture (P1045, Beyotime). The samples were fully homogenized using an electric homogenizer and placed on ice for full lysis. During the process, the sample was fully mixed every 10 min. The whole process was carried out in a low-temperature environment, lasting for 40 min. The fully lysed sample was centrifuged for 15 min (12,000 rpm, 4 °C), and the supernatant was collected and transferred to a new EP tube. Protein concentrations were determined using a bicinchoninic acid (BCA) protein assay kit (P0010, Beyotime).

An equal amount of protein sample was added to an AQTMPAGE preformed gel (AQ120-129) for electrophoresis (160 V, 30 min), and then the protein sample was transferred to a PVDF membrane (100 V, 60 min). The PVDF membrane was blocked with 5% skim milk powder at 37 °C for 2 h, and the PVDF membrane was blocked with primary antibody (4 °C, 12 h). The membranes were washed with Tris-buffered saline Tween-20 buffer (TBST) three times. The membrane was incubated with secondary antibody (RT, 1 h) and washed with TBST again (3 × 8 min, 60 rpm). The immunoblots were probed using western blot detection kits (BeyoECL Plus, P0018S, Beyotime, China) and visualized with an imaging system (Tanon-5200, Shanghai, China) [71].

### 4.10. Statistics

All experiments and data analyses were conducted by experimenters blinded to group assignment. Data are shown as the mean ± standard error of the mean (S.E.M.). One-way ANOVA, two-way ANOVA, two-way repeated-measures ANOVA, and the Kruskal–Wallis H test were performed using SPSS version 22 (IBM Analytics, Armonk, NY, USA). Details of each analysis are included in the figure legends and statistical significance was set at *p* < 0.05.

## 5. Conclusions

Overall, we explained in this study the importance of the downstream dopaminergic pathway from the A11 nucleus to SP5C for modulating migraine-related nociceptive sensitization and clarified the roles of D1DR and D2DR in SP5C in response to migraine stimulation and chronicity. The study findings contribute to the better construction of migraine-related neural circuits and elucidation of the pathogenesis of migraine (Figure 6).

## Figures and Tables

**Figure 1 ijms-24-16876-f001:**
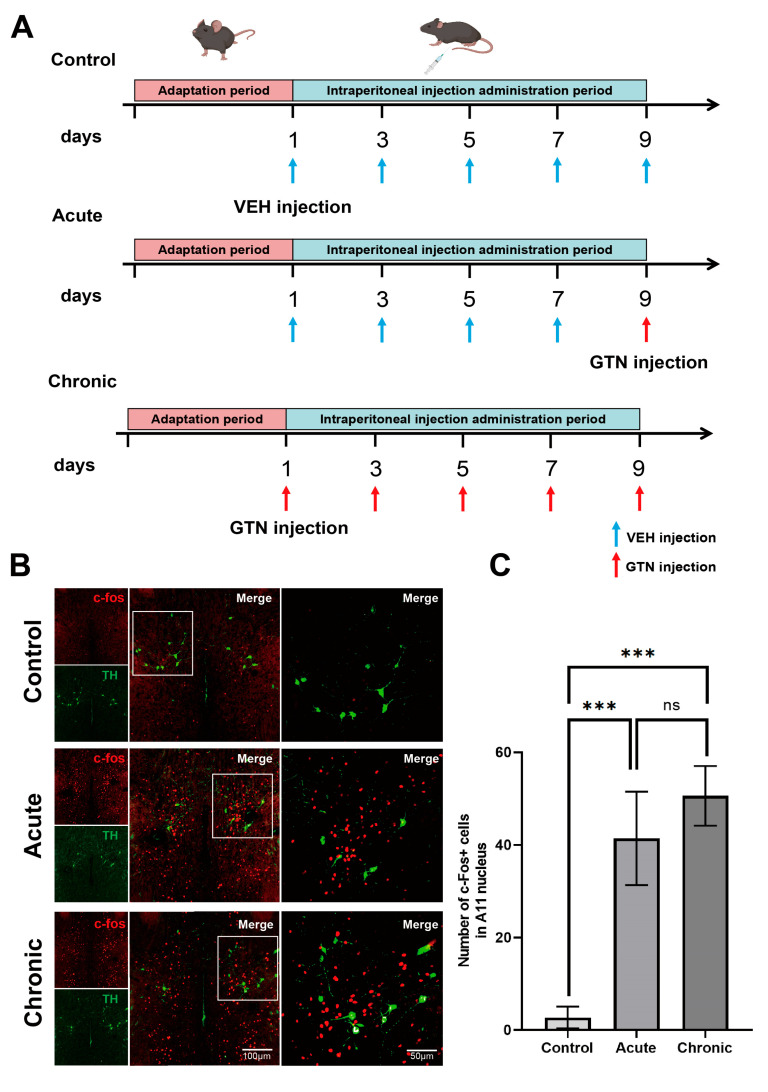
Phase I modeling and functional exploration of the A11 nucleus in the model. (**A**) Flow chart of the experiment. (**B**–**E**) Functional exploration of the A11 nucleus in the model and components. (**B**) The c-fos expression of A11 motifs in the control, acute, and chronic groups. TH (green) is used to indicate the position of A11 motifs, and the amount of c-fos (red) is used to characterize the activation degree of A11 motifs in each group, with a higher number representing a higher activation degree. (**C**) The number of c-fos+ cells in the A11 nucleus was counted in the three groups (control group: n = 10 mice per group, acute group: n = 9 mice per group, chronic group n = 8 mice per group) (**D**) Colocalization of VGAT+ (GABAergic neurons, red) neurons with c-fos+ (green) neurons in the A11 nucleus in the migraine model (acute vs. chronic groups). (**E**) Colocalization of TH+ (dopaminergic neurons, green) neurons with GABAA receptor+ (red) neurons in the A11 nucleus. (**F**) Differences in GAD (glutamic acid decarboxylase, used to characterize metabolic levels of GABAergic neurons) and TH (tyrosine hydroxylase, used to characterize metabolic levels of dopaminergic neurons) protein expression in the A11 nucleus among the three groups (n = 6 per group). Significance was assessed by Kruskal–Wallis H test with Mann–Whitney U post hoc comparison between groups (**C**; *** *p* < 0.001; ns, no significance) or by one-way ANOVA with post hoc comparison between groups (**F**; ns, no significance). All data are presented as the mean ± S.E.M.

**Figure 2 ijms-24-16876-f002:**
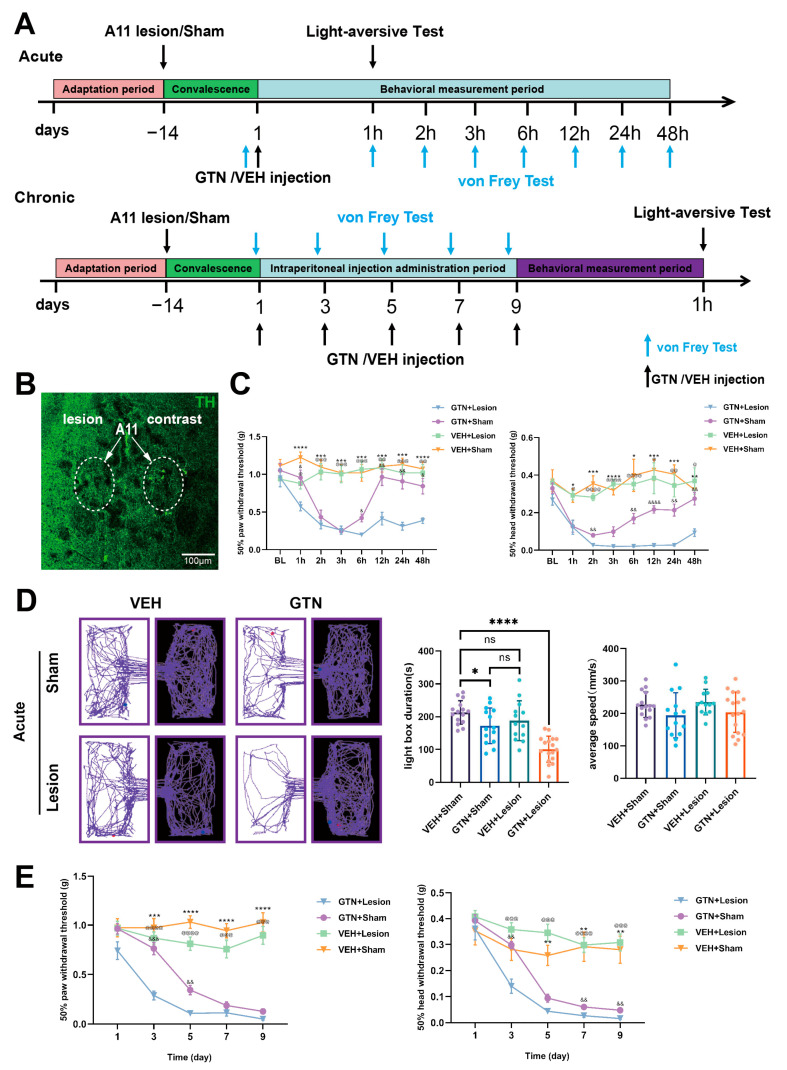
Phase II modeling and behavioral validation. (**A**) The GTN acute intervention group was treated with bilateral A11 nucleus dopaminergic neuron-specific injury or sham surgery after a brief acclimatization period. GTN or VEH was injected intraperitoneally 14 days after surgical recovery. The light and dark box test was performed 1 h after intraperitoneal injection, and the von Frey test was performed before and 1 h, 2 h, 3 h, 6 h, 12 h, 24 h, and 48 h after intraperitoneal injection (different batches of mice were used for the light and dark box test and the von Frey test). The GTN chronic intervention group had the same acclimatization and surgical and recovery periods as the acute group. The difference was that at the end of the recovery period, intraperitoneal injections of GTN or VEH were performed every other day for a total of 5 injections. The von Frey test was performed before each intraperitoneal injection. Photophobic behavioral tests were performed 1 h after the last intraperitoneal injection. (**B**) Staining plots of the unilateral A11 nucleus 14 days after 6-OHDA injection, with the number of TH+ (green) neurons used to characterize the extent of the damage. Unilateral injection was a convenient control, and all A11 damage in the experiment was bilateral. (**C**) Cephalic and facial pain thresholds and hind paw mechanical pain thresholds were measured in the GTN acute intervention group (n = 8 mice per group). & *p* < 0.05, && *p* < 0.01, &&&& *p* < 0.0001 GTN + lesion group vs. GTN + Sham group. @ *p* < 0.05, @@ *p* < 0.01, @@@ *p* < 0.001, @@@@ *p* < 0.0001 GTN + lesion group vs. VEH + lesion group. * *p* < 0.05, ** *p* < 0.01, *** *p* < 0.001, **** *p* < 0.0001 GTN + lesion group vs. VEH + Sham group. (**D**) Photophobic behavioral assays in the GTN acute intervention group (VEH + Sham: n = 15 mice per group; VEH + lesion: n = 13 mice per group; GTN + Sham: n = 14 mice per group; GTN + lesion: n = 17 mice per group). (**E**) Cephalofacial pain thresholds and hind paw mechanical pain thresholds were determined in the GTN chronic intervention group (n = 8 mice per group). && *p* < 0.01, &&& *p* < 0.001 GTN + lesion group vs. GTN + Sham group. @@@ *p* < 0.001, @@@@ *p* < 0.0001 GTN + lesion group vs. VEH + lesion group. ** *p* < 0.01, *** *p* < 0.001, **** *p* < 0.0001 GTN + Lesion group vs. VEH + Sham group. (**F**) Photophobic behavioral measurements in the GTN chronic intervention group (VEH + Sham: n = 13 mice per group; VEH + lesion: n = 12 mice per group; GTN + Sham: n = 8 mice per group; GTN + lesion: n = 9 mice per group). The significance of C, E was assessed by two-way repeated-measures with ANOVA Tukey’s multiple comparisons test between groups (**C** (**left**): main group effects: F (3, 28) = 88.82, *p* < 0.0001, main time effects: F (5.152, 144.3) = 8.232, *p* < 0.0001. Interaction between group and time: F (21, 196) = 4.006, *p* < 0.0001; **C** (**right**): main group effects: F (3, 28) = 64.24, *p* < 0.0001, main time effects: F (5.207, 145.8) = 6.211, *p* < 0.0001. Interaction between group and time: F (21, 196) = 2.324, *p* = 0.0014; E (left): main group effects: F (3, 28) = 82.01, *p* < 0.0001, main time effects: F (2.740, 76.71) = 30.95, *p* < 0.0001. Interaction between group and time: F (12, 112) = 9.329, *p* < 0.0001; E (right): main group effects: F (3, 28) = 30.92, *p* < 0.0001, main time effects: F (3.537, 99.05) = 40.66, *p* < 0.0001. Interaction between group and time: F (12, 112) = 5.992, *p* < 0.0001). Significance of D, F was assessed by one-way ANOVA with post hoc comparison between groups (**D**; * *p* < 0.05, **** *p* <0.0001; ns, no significance; F; ** *p* < 0.01, *** *p* < 0.001, **** *p* <0.0001; ns, no significance). All data are presented as the mean ± S.E.M. The mean ± S.E.M. is the mean ± S.E.M. of the post hoc comparison.

**Figure 3 ijms-24-16876-f003:**
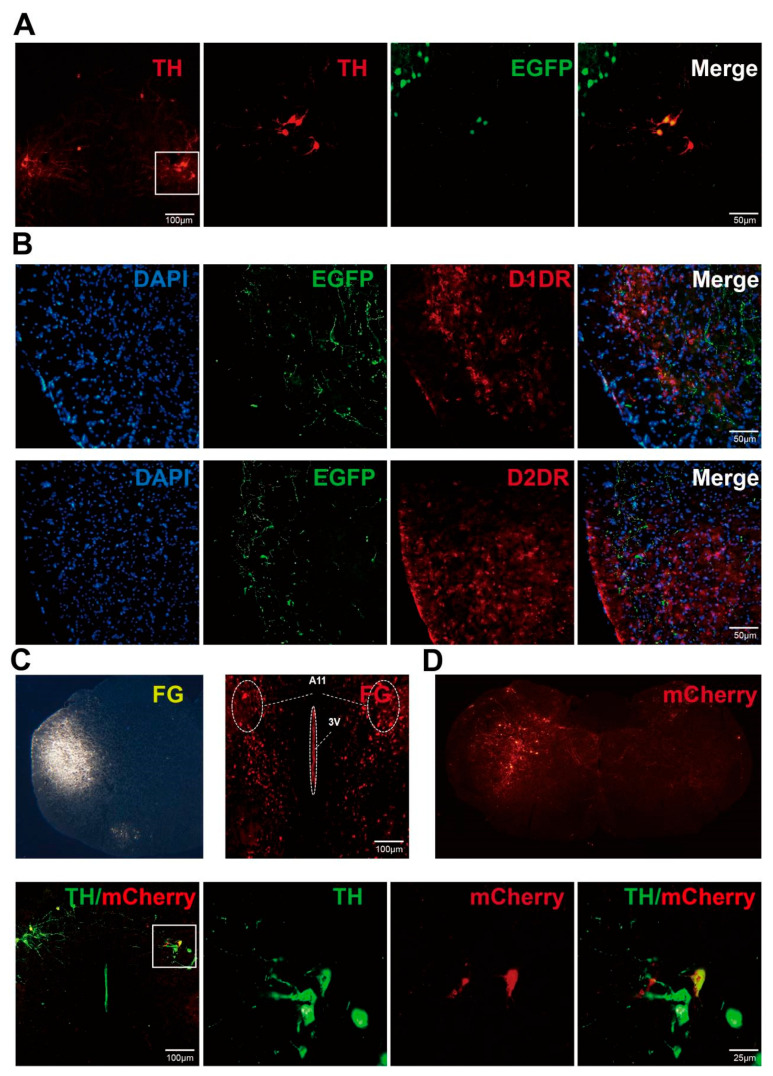
Paracrine and retrograde neuronal tracing of the A11 nucleus projected to SP5C. (**A**) Paracrine tracer virus pAAV-EF1-DIO-EGFP-WPRE was injected into the bilateral A11 nucleus of TH-cre mice and stained after 21 days of expression. TH+ (red) neurons were colocalized with pAAV-EF1-DIO-EGFP-WPR (green)-expressing neurons. (**B**) Fibers of pAAV-EF1-DIO-EGFP-WPRE projected to SP5C (green) colocalized with D1DR (red) and D2DR (red), respectively. (**C**) Fluorescent gold injected into SP5C (**left** panel) as well as FG+ (red) cytosolic bodies were found in the bilateral A11 nucleus. (**D**) Retrograde tracer viral pAAV-hSyn-DIO-mCherry-WPRE (red) injection sites in SP5C. pAAV-hSyn-DIO-mCherry-WPRE (red) projected to the bilateral A11 nucleus colocalized with TH+ (green) neurons.

**Figure 5 ijms-24-16876-f005:**
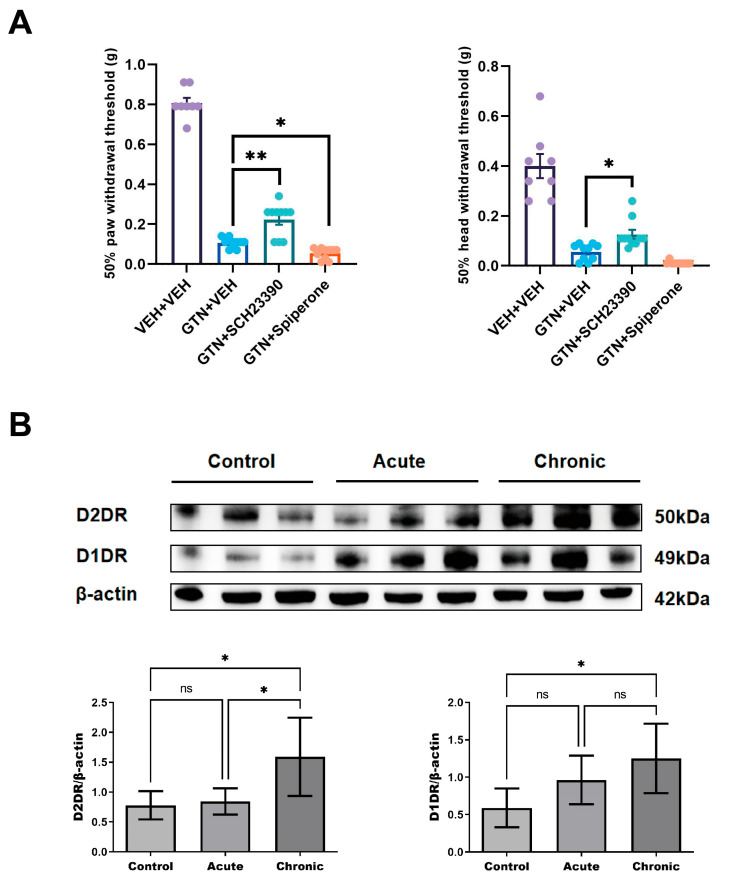
Pain behavioral measurements after modulation of dopamine receptors in SP5C and changes in dopamine receptor expression levels in different groups. (**A**) Behavioral changes after acute GTN intervention and administration of brain regions injected with the D1DR antagonist and D2DR antagonist (VEH + VEH: n = 8 mice per group, GTN + VEH: n = 10 mice per group, GTN + SCH23390: n = 10 mice per group; GTN + spiperone: n = 10 mice per group) (**B**) Changes in D1DR and D2DR protein expression levels in SP5C of the control, acute, and chronic stimulation groups (n = 6 mice per group). Significance was assessed by one-way ANOVA with post hoc comparison between groups (**A**, **B**; * *p* < 0.05, ** *p* < 0.01; ns, no significance). All data are presented as the mean ± S.E.M. The mean ± S.E.M. of the post hoc comparisons is shown in the table below.

**Figure 6 ijms-24-16876-f006:**
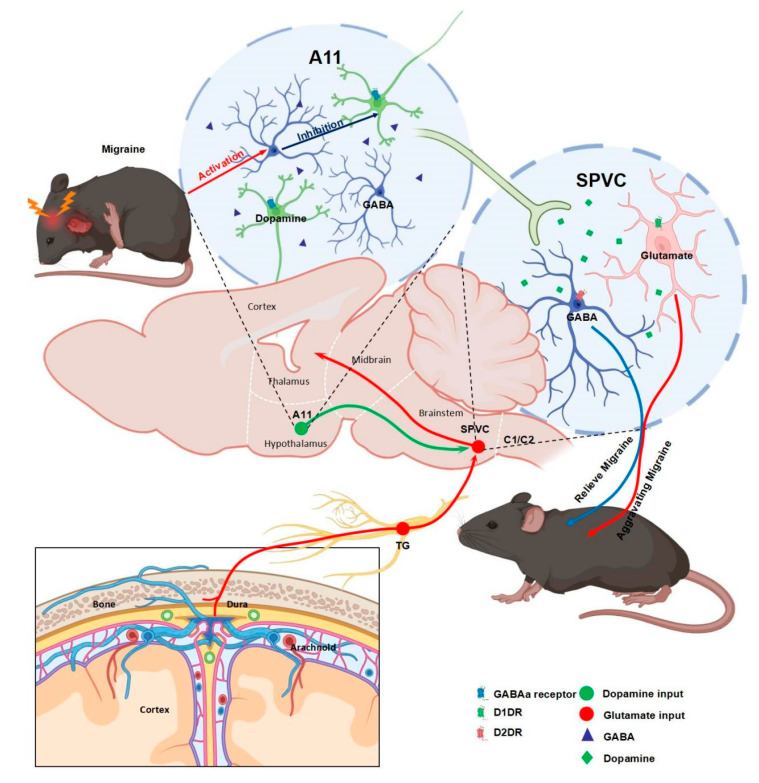
Schematic diagram of dopaminergic projections from the hypothalamic A11 nucleus to the spinal trigeminal nucleus involved in bidirectional modulation of migraine headache.

**Table 1 ijms-24-16876-t001:** Key resources table.

Antibody	Dilution	Vendor	Catalog Number
Anti-Tyrosine Hydroxylase	1:1000	GeneTex	GTX10372
Anti-c-fos	1:1000	CST	2250S
Anti-DRD1A	1:500	Millipore	MAB5290
Anti-DRD2	1:500	Millipore	AB5084P
Anti-GABA_A_ receptor	1:400	Abcam	ab300069
Anti-GAD65	1:400	Abcam	ab239372
Anti-DRD1	1:500	Abcam	ab279713
Anti-DRD2	1:500	Proteintech	55084
Anti-vGluT2	1:1000	Abcam	ab79157
Anti-beta Actin	1:1000	Proteintech	20536
Goat anti-mouse 488	1:1000	Abcam	ab150113
Goat anti-rabbit 594	1:1000	Abcam	ab150080
HRP- Goat Anti-Mouse lgG	1:1000	Proteintech	SA00001-1
HRP- Goat Anti-Rabbit lgG	1:1000	Proteintech	SA00001-2

## Data Availability

Data is contained within the article and supplementary material. All data, reagents, resources, and protocols are available from the corresponding author upon reasonable requests.

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
