# Peer review of "Dopaminergic Projections from the Hypothalamic A11 Nucleus to the Spinal Trigeminal Nucleus Are Involved in Bidirectional Migraine Modulation"

_ijms, 2023, doi:10.3390/ijms242316876_

Round 1

Reviewer 1 Report

Comments and Suggestions for Authors

ijms-2672900-peer-review-v1

The manuscript presented by Chenhao Li et al. titled: “Dopaminergic projections from hypothalamic A11 nucleus to the spinal trigeminal nucleus are involved in bidirectional migraine modulation “analytical cross-sectional study” is well-written, clear, and easy to read. The topic is interesting; therefore, it adds clustered information to the subject area of neuroinflammation.

Said that, first of all, take off the sections (1) Background; (2) Method; 3 Results, and (4) Discussion in the abstract. Even, do not use acronyms in the abstract. Put materials and methods after the introduction; there are acronyms useful for the text

It seems that chronic and acute phases of migraine gave same response. I think that the dynamics of migration should be considered in the discussion, as well as epigenomic change induced by this condition. So please introduce small noncoding RNA (several are conserved amongst species) into the discussion. See reference

doi: 10.1093/ijnp/pyv025;

https://doi.org/10.1080/21691401.2019.1703729;

doi: 10.2174/2211536606666170913152821;

doi: 10.1186/s12974-021-02342-5.

In real life, the migraine attack starts from pediatric/adolescence. See reference

DOI: 10.1212/CON.0000000000000993,

doi: 10.1016/j.ncl.2019.07.009. 

Please discuss this point as well. The mice model is 20-30 g; please also indicate the age in terms of weeks.

Author Response

Dear Editors and Reviewers,

Thank you for your letter and for the reviewers’ comments concerning our manuscript entitled “Dopaminergic projections from hypothalamic A11 nucleus to the spinal trigeminal nucleus are involved in bidirectional migraine modulation” (Manuscript ID:ijms-2672900).

These comments are all valuable and very helpful for revising and improving our paper, as well as the important guiding significance to our research. We have studied the comments carefully and have made corrections which we hope meet with approval. We use the ‘tracked changes’ function in Word Software to track all the revisions in the ‘Change-tracked revised manuscript’. We also submit a neat edition of the manuscript. We sincerely hope that this revised manuscript and point-by-point response letter to the comments have addressed all your comments and suggestions. The point-by-point response letter is detailed below. Once again, thank you and the reviewers very much for your comments and suggestions.

Yours sincerely,

Shengyuan Yu

Responses to the reviewers’ comments

Reviewer #1:The manuscript presented by Chenhao Li et al. titled: “Dopaminergic projections from hypothalamic A11 nucleus to the spinal trigeminal nucleus are involved in bidirectional migraine modulation “analytical cross-sectional study” is well-written, clear, and easy to read. The topic is interesting; therefore, it adds clustered information to the subject area of neuroinflammation.

  • Said that, first of all, take off the sections (1) Background; (2) Method; 3 Results, and (4) Discussion in the abstract. Even, do not use acronyms in the abstract. Put materials and methods after the introduction; there are acronyms useful for the text.

Response: Thank you very much for the insightful questions and comments. I have removed the sections from the abstract as requested.In terms of ordering the sections of the paper, I followed the article template downloaded from the MDPI website.And I browsed through the recent articles published in IJMS, and the MATERIALS and METHODS are placed at the end of the article (Change-tracked revised manuscript).

  • It seems that chronic and acute phases of migraine gave same response. I think that the dynamics of migration should be considered in the discussion, as well as epigenomic change induced by this condition. So please introduce small noncoding RNA (several are conserved amongst species) into the discussion. See reference

doi: 10.1093/ijnp/pyv025;

https://doi.org/10.1080/21691401.2019.1703729;

doi: 10.2174/2211536606666170913152821;

doi: 10.1186/s12974-021-02342-5.

In real life, the migraine attack starts from pediatric/adolescence. See reference

DOI: 10.1212/CON.0000000000000993,

doi: 10.1016/j.ncl.2019.07.009.

Response: Thank you so much for your advice. I have carefully read the references you provided and introduced small noncoding RNA in the discussion section.(See line 424 on page 19, Change-tracked revised manuscript).

3)Please discuss this point as well. The mice model is 20-30 g; please also indicate the age in terms of weeks.

Response: Thank you very much for the constructive comments.I have inserted information about the weekly age of the mice into the article (See line 446 on page 19, Change-tracked revised manuscript).

Special thanks to you for your good comments. It’s our honor to have our paper reviewed by you, thank you for your hard work and professionalism, you raised questions on even a tiny mistake, it must have taken you a long time, your valuable suggestions will make our paper better, and your attitude toward work will make a positive effect to us and our further studies.

We tried our best to improve the manuscript and made some changes in the manuscript.  These changes will not influence the framework of the paper. We appreciate your warm work earnestly and hope that the correction will meet with approval.

Once again, thank you very much for your comments and suggestions.

Reviewer 2 Report

Comments and Suggestions for Authors

This is an interesting study reporting dopaminergic projections from the A11 hypothalamic nucleus to GABAergic neurons in trigeminal nucleus caudalis or SP5C playing a critical role in migraine-relevant nociceptive processing. As the role of A11 spinal projections in pain modulation was previously established, dopaminergic regulation of SP5C inhibitory interneurons found in this study is novel and may be a potential target for pain mitigation. However, I had some major concerns regarding key experimental controls, data processing, clarity of data presentation, and attention to detail while writing, which are listed below.

Major issues

1)    The study lacks physiological evidence showing dopaminergic neurons in the A11 hypothalamic nucleus drive GABAergic neuron activity in the trigeminal nucleus caudalis. Optogenetic stimulation of A11 DA afferents while recording activity in SP5C GABAergic neurons, or a similar experiment, may provide direct physiological evidence and improve the impact of these findings.

2)    Beyond pain modulation, dopaminergic neurons form A11 project to several regions in the spinal cord and are implicated in the regulation of locomotion and restless leg syndrome. The authors use chemical lesions to abate DA neurons in A11. This may cause potential defects in the locomotion, that should be addressed by behavioral assays like open field tests, rotarod, and gait tests. 

3)    In Figure 1, a zoomed-out image of an entire slice showing the anatomical location of A11 nuclei is recommended.  Currently, it is hard to see a difference in VGAT immunoreactivity in A11 nuclei compared to neighboring regions in the image in panel D. Also in the schematic, the authors should specify when the tissue was collected for histology.  

4)    Authors only show qualitative data for localization, in the form of representative figures. The colocalized puncta should be quantified. The representative images in Figure 1E suggest that GABA-a receptors show 100% colocalization with TH+ neurons, which seems quite odd. Authors should confirm this by electrophysiology, or at least testing if GABA-A receptors are expressed on GABAergic interneurons (vgat+ cells) in A11 nucleus. Quantification of colocalized puncta might help with this too.

5)    Authors should specify what “n” means for each experiment (e.g. sections or animals). Authors should also specify how many sections were collected per animal?

6)     In Figure 2B, the hemisphere with the lesion has more TH immunoreactivity in the neuropil, compared to the control side. Quantification of TH+ neurons in lesion vs control side might be more convincing. Also, a zoomed-out image of an entire slice showing the region specificity of 6-OHDA injection is recommended. This will convince the reviewers that TH+ neurons in other dopaminergic nuclei were left intact.

7)    A Vital control (vehicle+Sham) is missing from von Fray data in Figure 2. Including this control group is critical as a comparison with the vehicle+A11 lesion group informs if an A11 lesion alone has a direct effect on tactile sensitivity. For chronic GTN injections, authors fail to adequately describe when the animals were tested von-fray presented in panel E. Authors should also specify if the plots represent the mean responses after all 9 trials or just the single trial. As a control, the authors should also include input-output curves for withdrawal responses with different pressures of von Fray filament.

8)    In Figure 2C, all groups have an n=6, except for GTN+lesion group which has n=15. This difference is quite significant. Increasing the sample size may artificially increase the proportion of statistically significant comparisons. The authors should specify if there’s a logical reason behind this difference.

9)    For Figure 3, representative images are not adequate to convince the reviewers that the expression of D1/2DRs was specific to SP5C. A zoomed-out image showing subdivisions of mouse TCN (oralis, interpolar and caudalis) may be necessary.

10) For anterograde tracing experiments, the authors performed bilateral stereotactical injections. Why not unilateral injections consistent with the unilateral lesions? The contralateral side after unilateral injections can be an effective control. Also, it would be interesting to see if A11 dopaminergic neurons project bilaterally to the trigeminal nucleus caudalis.

11) For tracing experiments, the authors wrote “Paracrine 265 tracer virus pAAV-EF1-DIO-EGFP-WPREwas used.  This merely indicates the plasmid, not the actual AAV. The authors must mention the viral serotypes used for anterograde and retrograde tracing. Also, the authors should explain the choice of EF1 promoter. Ideally, authors should have used DA neuron-specific mTH promoter (addgene 99128), if not a pan-neuronal promoter like Syn that they used for retrograde tracing.

12) The authors do not include western blots in Figure 1 and present cut-out bands in Figure 5. Full blots with visible protein ladders are recommended.

13) In Figure 5, the authors show von Fray tests with dopamine receptor D1/2 DR antagonists but fail to mention when they were administered. The authors should clarify this.  Perhaps include a schematic like Figure 1?. 

14) Statistics: Did the authors assume normality of the distribution? Authors should test for normality and then use tests designed for comparing either parametric or non-parametric data should be used.

Minor issues:

15) The figures are organized poorly and sometimes the individual panels are seemingly arranged without a logical flow. Some panels and graphs are missing their subheadings.

16) There are a few tense agreement issues in the manuscript along with a lack of clarity while describing background information at several places.

17) Appropriate citations are missing in several places. for e.g., in the introduction (lines 60-62): “In addition to regulating body temperature, sleep, and food and water intake, the hypothalamus has specific control over the autonomic nervous system and cyclical phenomena (e.g., circadian and diurnal rhythms)”, requires several appropriate citations.

Due to the sheer number of major concerns, I recommend that the manuscript should be rejected in its current format.

Comments on the Quality of English Language

1)    There are a few tense agreement issues in the manuscript along with a lack of clarity while describing background information at several places.

Author Response

Dear Editors and Reviewers,

Thank you for your letter and for the reviewers’ comments concerning our manuscript entitled “Dopaminergic projections from hypothalamic A11 nucleus to the spinal trigeminal nucleus are involved in bidirectional migraine modulation” (Manuscript ID:ijms-2672900).

These comments are all valuable and very helpful for revising and improving our paper, as well as the important guiding significance to our research. We have studied the comments carefully and have made corrections which we hope meet with approval. We use the ‘tracked changes’ function in Word Software to track all the revisions in the ‘Change-tracked revised manuscript’. We also submit a neat edition of the manuscript. We sincerely hope that this revised manuscript and point-by-point response letter to the comments have addressed all your comments and suggestions. The point-by-point response letter is detailed below. Once again, thank you and the reviewers very much for your comments and suggestions.

Yours sincerely,

Shengyuan Yu

Responses to the reviewers’ comments

Reviewer #2:This is an interesting study reporting dopaminergic projections from the A11 hypothalamic nucleus to GABAergic neurons in trigeminal nucleus caudalis or SP5C playing a critical role in migraine-relevant nociceptive processing. As the role of A11 spinal projections in pain modulation was previously established, dopaminergic regulation of SP5C inhibitory interneurons found in this study is novel and may be a potential target for pain mitigation. However, I had some major concerns regarding key experimental controls, data processing, clarity of data presentation, and attention to detail while writing, which are listed below.

Major issues

  •   The study lacks physiological evidence showing dopaminergic neurons in the A11 hypothalamic nucleus drive GABAergic neuron activity in the trigeminal nucleus caudalis. Optogenetic stimulation of A11 DA afferents while recording activity in SP5C GABAergic neurons, or a similar experiment, may provide direct physiological evidence and improve the impact of these findings.

Response:Thank you very much for the insightful questions and comments.I hold the same opinion as you and I am already working on this part in my follow-up experiment. I hope to make exploring the direct connection between the DAergic inputs of A11 and the GABAergic neurons of SP5C as well as glutamatergic neurons a centerpiece of my next work for future publication.

  • Beyond pain modulation, dopaminergic neurons form A11 project to several regions in the spinal cord and are implicated in the regulation of locomotion and restless leg syndrome. The authors use chemical lesions to abate DA neurons in A11. This may cause potential defects in the locomotion, that should be addressed by behavioral assays like open field tests, rotarod, and gait tests.

Response:Thank you very much for the insightful questions and comments. Based on previous studies, the A11 nucleus is likely to be involved in the pathophysiology of restless legs syndrome. In several experiments, mice with restless legs syndrome were modeled using 6-OHDA to specifically damage DAergic neurons in the A11 nucleus. In this model, the mice showed only an increase in involuntary leg movements during sleep, but were not significantly affected during the waking state.See reference

DOI: 10.1002/mds.24058

DOI: 10.1097/nen.0b013e3180517b5f

When we performed behavioral measurements on the mice, the mice were all in an awake state. In the light and dark box experiments, we recorded the average movement speed (See figure2 D,F on page 9, Change-tracked revised manuscript) to indirectly reflect the mice's locomotor ability. From the recorded results, the destruction of the A11 nucleus did not significantly affect the motor function of the mice in the awake state. In this experiment, we mainly investigated the presence of migraine headache and did not consider more detailed motor behavioral tests for the time being. In the next further work, we will further improve the related behavioral science.

  • In Figure 1, a zoomed-out image of an entire slice showing the anatomical location of A11 nuclei is recommended.  Currently, it is hard to see a difference in VGAT immunoreactivity in A11 nuclei compared to neighboring regions in the image in panel D. Also in the schematic, the authors should specify when the tissue was collected for histology.  

Response:Thank you very much for the constructive comments. I have made the required adjustments to the D in Figure 1, in which we can find aggregates of VGAT+ neurons in the region of the A11 nucleus pulposus (classical location: paraventricular third ventricle) and co-localized with c-fos+ neurons. These manifestations are significantly different from the surrounding tissue(See figure1 D on page 7, Change-tracked revised manuscript).

4)    Authors only show qualitative data for localization, in the form of representative figures. The colocalized puncta should be quantified. The representative images in Figure 1E suggest that GABA-a receptors show 100% colocalization with TH+ neurons, which seems quite odd. Authors should confirm this by electrophysiology, or at least testing if GABA-A receptors are expressed on GABAergic interneurons (vgat+ cells) in A11 nucleus. Quantification of colocalized puncta might help with this too.

Response:Thank you very much for the insightful questions and comments. After several immunohistochemical stainings, we found that GABAA receptor+ neurons did co-localize with TH+ neurons in bilateral A11 nuclei, and this finding was relatively stable. Co-localization staining of GABAA receptors with vgat could not be carried out for the time being due to the lack of vgat-Tdtomato mice. Immunofluorescence staining with specific antibodies in the A11 nucleus was difficult to render the morphology of GABA neurons. I will address these issues in future studies.

5)    Authors should specify what “n” means for each experiment (e.g. sections or animals). Authors should also specify how many sections were collected per animal?

Response:Thank you very much for the constructive comments. I have labeled each“n”in the figure legends. For counting, we ensured that at least 3 mice per group were collected with 2-3 (subject to the small size of the A11 nucleus) classically located, well-morphologized sections per mouse (Change-tracked revised manuscript).

  • In Figure 2B, the hemisphere with the lesion has more TH immunoreactivity in the neuropil, compared to the control side. Quantification of TH+ neurons in lesion vs control side might be more convincing. Also, a zoomed-out image of an entire slice showing the region specificity of 6-OHDA injection is recommended. This will convince the reviewers that TH+ neurons in other dopaminergic nuclei were left intact.

Response:Thank you very much for your suggestions.The destruction of bilateral A11 nucleus using 6-OHDA is a more established technique. The unilateral destruction of A11 nucleus DA neurons is shown in Figure 2 in order to compare left and right to visualize the effect of the destruction more intuitively. In the pre-replacement image, the fluorescence intensity on the control side appears to be dimmer than on the damaged side, due to the fact that the control side of that image was subjected to confocal scanning prior to taking the picture and the fluorescence intensity decayed. I have replaced the diagram with a more visual one as requested, and you can see that there are almost no TH+ neurons present in the A11 nucleus on the damaged side(See figure2 B on page 9, Change-tracked revised manuscript).

7)    A Vital control (vehicle+Sham) is missing from von Fray data in Figure 2. Including this control group is critical as a comparison with the vehicle+A11 lesion group informs if an A11 lesion alone has a direct effect on tactile sensitivity. For chronic GTN injections, authors fail to adequately describe when the animals were tested von-fray presented in panel E. Authors should also specify if the plots represent the mean responses after all 9 trials or just the single trial. As a control, the authors should also include input-output curves for withdrawal responses with different pressures of von Fray filament.

Response:Thank you for correcting the errors in the article.Vehicle+Sham's behavioral stats we have been recording and calculating were not showing up during image processing and inspection. The behavioral statistics graphs have been replaced accordingly as requested (See figure2 C,E on page 9, Change-tracked revised manuscript).

Regarding the calculation of the value of the von fray, we have used the UP-DOWN recording method, which is widely used in most of the current pain behavioral measurements. We have added details about the calculation method in the Materials and Methods section (See line 454  on page 19, Change-tracked revised manuscript).

8)    In Figure 2C, all groups have an n=6, except for GTN+lesion group which has n=15. This difference is quite significant. Increasing the sample size may artificially increase the proportion of statistically significant comparisons. The authors should specify if there’s a logical reason behind this difference.

Response:Thank you very much for the insightful questions and comments. This is because the cephalic and facial pain thresholds of GTN+lesion fluctuated more during the preexperiment, and a small number of mice would be in such poor condition after GTN injection that they could not continue to complete the behavioral assays. Therefore, in each round of behavioral assay during the formal experiment, the number of GTN+lesion group was higher relative to the other three groups. When the subsequent data were summarized and counted, the GTN+lesion group was then included in the statistics in a relatively larger number as well.

9)    For Figure 3, representative images are not adequate to convince the reviewers that the expression of D1/2DRs was specific to SP5C. A zoomed-out image showing subdivisions of mouse TCN (oralis, interpolar and caudalis) may be necessary.

Response:Thank you very much for the constructive comments. As you say, D1DR and D2DR are not specific to the SP5C, and dopamine receptors are widely distributed in multiple brain regions throughout the brain. However, in our behavioral assay, we used the von-fray to determine whether migraine mice showed cephalic-facial skin nociceptive sensitization, and the secondary neurons innervating cephalic-facial sensation are mainly distributed in the SP5C region, so we focused only on DA receptors in the SP5C region, including in the subsequent pharmacological modulation, and we placed the administration cannulae bilaterally in the SP5C region. Of course, whether DA receptors in other TNC regions (oralis and interpolar) may also be involved in the regulation of migraine is still unknown, and we will further explore this in future research work, which I believe will be a very meaningful study.

10) For anterograde tracing experiments, the authors performed bilateral stereotactical injections. Why not unilateral injections consistent with the unilateral lesions? The contralateral side after unilateral injections can be an effective control. Also, it would be interesting to see if A11 dopaminergic neurons project bilaterally to the trigeminal nucleus caudalis.

Response:Thank you very much for the insightful questions and comments. During the experiment, because of the small volume of A11 nuclei, the amount of injected virus is much smaller than normal nuclei, and A11 nuclei are very close to the midline, the success rate of unilateral injection is lower. Moreover, to ensure the specificity of the viral tracer, our tracer viruses were used in combination with TH-cre mice, which are expensive. In order to improve the success rate of tracing and avoid the waste of resources, we used bilateral injection for the downstream downstream tracing of A11. Whereas, during retrograde tracing, the injection sites were located bilaterally at SP5C, and the injection area was large compared to that of the A11 nucleolus. However, because SP5C is located in the medulla oblongata of mice, in which there are more centers related to life activities, if we choose bilateral injection, the pressure in the medulla oblongata is high, the needle path passes through many anatomical structures, the anesthesia time is long, and the mortality rate of mice is high. Because we use unilateral injection for retrograde tracing. Your suggestion is of very far-reaching significance, and we will follow up with a large number of experiments to explore this interesting projection method.

11) For tracing experiments, the authors wrote “Paracrine 265 tracer virus pAAV-EF1-DIO-EGFP-WPRE“ was used.  This merely indicates the plasmid, not the actual AAV. The authors must mention the viral serotypes used for anterograde and retrograde tracing. Also, the authors should explain the choice of EF1 promoter. Ideally, authors should have used DA neuron-specific mTH promoter (addgene 99128), if not a pan-neuronal promoter like Syn that they used for retrograde tracing.

Response:Thank you very much for the constructive advice. The serotypes of the anterograde tracingviruses we used were AAV9 and the serotypes of the retrograde tracing viruses were Retro. we used anterograde and retrograde tracing viruses with DIO sequences in combination with TH-cre mice, and this pairing pattern was superior in stability and specificity to the dopamine neuron-specific promoter (Change-tracked revised manuscript).

12) The authors do not include western blots in Figure 1 and present cut-out bands in Figure 5. Full blots with visible protein ladders are recommended.

Response:Thank you very much for the insightful questions and comments. I have added protein imprints to the Figure 1F as requested (See figure1 F on page 7, Change-tracked revised manuscript). And the cut strips in Figure 1 and Figure 5 are just the excess blanks cut off. This is because during the experiment, I was concerned about the tendency to damage the proteins when over-cutting the PVDF membrane, so I kept the larger membrane when incubating the antibody.

13) In Figure 5, the authors show von Fray tests with dopamine receptor D1/2 DR antagonists but fail to mention when they were administered. The authors should clarify this.  Perhaps include a schematic like Figure 1?.

Response:Thank you very much for the constructive advice. Cannula administration is usually performed 30-40 min prior to behavioral measurements.I have added it to the Materials and Methods section (See line 497 on page 20, Change-tracked revised manuscript).

14) Statistics: Did the authors assume normality of the distribution? Authors should test for normality and then use tests designed for comparing either parametric or non-parametric data should be used.

Response:Thank you very much for the constructive advice.We tested the data for normality. Data that did not fit the normal distribution, we tested hypotheses using the Kruskal-Wallis H test.

Minor issues:

15) The figures are organized poorly and sometimes the individual panels are seemingly arranged without a logical flow. Some panels and graphs are missing their subheadings.

Response:Thank you very much for your suggestions. I have made typographical changes to the figures to ensure the wholeness of each one (Change-tracked revised manuscript).

16) There are a few tense agreement issues in the manuscript along with a lack of clarity while describing background information at several places.

Response:Thank you very much for your suggestions.I improved the English style of this article through American Journal Experts to make the expressions in the article more accurate. 

17) Appropriate citations are missing in several places. for e.g., in the introduction (lines 60-62): “In addition to regulating body temperature, sleep, and food and water intake, the hypothalamus has specific control over the autonomic nervous system and cyclical phenomena (e.g., circadian and diurnal rhythms)”, requires several appropriate citations.

Response:Thank you very much for your suggestions.This part of the description of the context is not highly relevant to the content of this study, and I removed this part of the contextual description due to a lack of reference download permissions during the insertion of the relevant references(Change-tracked revised manuscript).

Special thanks to you for your good comments. It’s our honor to have our paper reviewed by you, thank you for your hard work and professionalism, you raised questions on even a tiny mistake, it must have taken you a long time, your valuable suggestions will make our paper better, and your attitude toward work will make a positive effect to us and our further studies.

We tried our best to improve the manuscript and made some changes in the manuscript.  These changes will not influence the framework of the paper. We appreciate your warm work earnestly and hope that the correction will meet with approval.

Once again, thank you very much for your comments and suggestions.

Round 2

Reviewer 2 Report

Comments and Suggestions for Authors

Comments on the Quality of English Language

minor improvements in the writing will be beneficial. 

Author Response

Dear Editors and Reviewers,

Thank you for your letter and for the reviewers’ comments concerning our manuscript entitled “Dopaminergic projections from hypothalamic A11 nucleus to the spinal trigeminal nucleus are involved in bidirectional migraine modulation” (Manuscript ID:ijms-2672900).

These comments are all valuable and very helpful for revising and improving our paper, as well as the important guiding significance to our research. We have studied the comments carefully and have made corrections which we hope meet with approval. We use the ‘tracked changes’ function in Word Software to track all the revisions in the ‘Change-tracked revised manuscript’. We also submit a neat edition of the manuscript. We sincerely hope that this revised manuscript and point-by-point response letter to the comments have addressed all your comments and suggestions. The point-by-point response letter is detailed below. Once again, thank you and the reviewers very much for your comments and suggestions.

Yours sincerely,

Shengyuan Yu

Responses to the reviewers’ comments

Reviewer #2:The revised manuscript has significant improvements and authors adequately addressed most of the reviewers’comments. Yet, there are some key issues outlined below that authors need to address.

  1. The authors did not provide quantification of the colocalized puncta (Forg., X% of total GABAaR punctacolocalized with TH+ signal). Without it the IHC colocalization data is merely qualitative.

Response:Thank you very much for your suggestions.I have related the statistical results of co-localization of TH+ neurons with GABAA+ neurons in figure1S (supplementary material). After the statistics we found that the distribution of GABAA receptors on the surface of DA neurons was more prevalent in the single GTN stimulation group and in the repeated GTN stimulation group compared to the control group. And interestingly, compared to other neurons, DAergic neurons in the A11 nucleus showed an enrichment of GABAA receptors on the surface after nitroglycerin intervention (See figure1S A,B on page 24, Change-tracked revised manuscript), which was different from other GABAA+ cells in the nucleus. Neurons in the cortex are better sensitized to various antibodies and are more likely to show morphology compared to neurons in subcortical nuclei. We therefore captured images of GABAA+ cells in the cortex as a comparison. Positive immunoreactivity was stronger in GABAA+ DAergic neurons in the A11 nucleus compared to GABAA+ cells in the cortex (See figure1S C on page 24, Change-tracked revised manuscript). This interesting phenomenon deserves to be pursued in subsequent studies.

  1. Authors should specify information about the antibodies used in the study like vendor and catalog no. This is critical to ensure reproducibility of the data.

Response:Thank you very much for the insightful questions and comments.I have presented information about the antibodies used in the study in the manuscript as requested (See line 581 on page 22, Change-tracked revised manuscript).

  1. At the reviewer’s request, the authors do provide full images of the blots with the revised Yet, none of the blots show protein ladder for western blots. Including protein ladders ensures that the band is specific to the protein of interest based on their molecular weight.

Response:Thank you very much for your suggestions. I've added the protein ladder to the appropriate section of the manuscript (See figure1 F on page 8 and figure5 B on page 16, Change-tracked revised manuscript).

  1. While addressing reviewers comment regarding a large difference in the N’s in von-fray test, theauthors mentioned that some of the animals in GTN-lesion group that are included in the analysis, were in poor health and could not complete the task. Therefore, the authors had to add more animals to ensure sufficient measurements at everytimepoint. This is very concerning, as poor health of animals disqualifies the animals from participating in the experiments. Moreover, regulatory bodies mandate that the animals should be euthanized if any perturbation / surgical procedure causes severe pain or health issues. The authors should exclude data collected from all animals that were in “poor health” at the time of the experiment.

Response:Thank you very much for the constructive comments.I think there was some misunderstanding when the explanation was first made. In the pre-experimental phase, there will be some animals in the GTN+Lesion group that will not be able to cooperate in completing the behavioral studies within 1h-2h after the GTN injection because of severe pain. After anticipating this result, we appropriately expanded the number of experimental animals in the GTN+Lesion group during the formal experiment. However, this situation basically did not occur during the formal experiment. All behavioral data included in the statistics were valid and reliable. Since there were cases of unequal sample sizes between groups in the data statistics of von fray test, we supplemented and recounted the data (See figure2 C,E on page 10, Change-tracked revised manuscript).

  1. In figure 2, the authors used one-way ANOVA and Kruskal Wallis tests for 2C and 2E respectively. In this case, the authors should have used two-way ANOVA, as they are comparing measurementsfrom 4 groups across 5 different timepoints. Also, authors should mention which post-hoc tests they used for multiple comparisons. Additionally, two way ANOVA should be used to analyze data in Figure 5A, which compares the effects of D1/D2 R antagonists across vehicle and GTN groups.

Response:Thank you very much for the insightful questions and comments.I have changed the statistics in Figures 2C and E as requested, using two-way ANOVA on the data from the von-frey test (See line 255 on page 11, Change-tracked revised manuscript). The discussion about the statistical methods of Figure 5A is as follows: since I viewed the intraperitoneal injection and SP5C cannula administration (e.g., GTN+Spiperone) as a whole as one treatment, a one-way ANOVA was more appropriate for the data in this part. Because we did not establish groups of VEH+D1/D2 R antagonists during the experiment, a two-way ANOVA was not appropriate. This part of the comparison centers on the effect of D1/D2 R antagonists on the pain threshold of mice with GTN intervention. The VEH+VEH group was established mainly to exclude the effect of tube placement surgery on the pain threshold of mice (See figure5 A on page 16, Change-tracked revised manuscript).

  1. Figure 1: E and F should be flipped.

Response:Thank you very much for your suggestions.I've flipped the positions of E and F in Figure 1 (See figure1 E, F on page 8, Change-tracked revised manuscript).

Special thanks to you for your good comments. It’s our honor to have our paper reviewed by you, thank you for your hard work and professionalism, you raised questions on even a tiny mistake, it must have taken you a long time, your valuable suggestions will make our paper better, and your attitude toward work will make a positive effect to us and our further studies.

We tried our best to improve the manuscript and made some changes in the manuscript.  These changes will not influence the framework of the paper. We appreciate your warm work earnestly and hope that the correction will meet with approval.

Once again, thank you very much for your comments and suggestions.
